# The Elbrus (Caucasus, Russia) ice core record - Part 2: history of desert dust deposition

Stanislav Kutuzov[1], Michel Legrand[2,3], Suzanne Preunkert[2,3], Patrick Ginot[2,3,6], Vladimir Mikhalenko[1], Karim Shukurov[4], Alexey Poliukhov[5], Pavel Toropov[1,5]

[1] Institute of Geography, Russian Academy of Sciences, Moscow, 119017, Russia
[2] Université Grenoble Alpes, CNRS, Institut des Géosciences de l'Environnement (IGE), Grenoble, 38402, France
[3] CNRS, Institut des Géosciences de l'Environnement (IGE), Grenoble, 38402, France
[4] A.M. Obukhov Institute of Atmospheric Physics of Russian Academy of Sciences, Moscow, 119017, Russia
[5] Faculty of Geography, Lomonosov Moscow State University, Moscow, 119991, Russia
[6] Observatoire des Sciences de l'Univers de Grenoble, IRD/UGA/CNRS, Grenoble, 38400, France

*Correspondence to*: Stanislav Kutuzov (kutuzov@igras.ru)

**Abstract.** Ice cores are one of the most valuable paleoarchives. Records from ice cores provide information not only about the amount of dust in the atmosphere, but also about dust sources and their changes in the past. In 2009, a 182 m long ice core was recovered from the western plateau of Mt. Elbrus (5115 m a.s.l.). This record was further extended after a shallow ice core was drilled in 2013. Here we analyse $Ca^{2+}$ concentrations, a commonly used proxy of dust, recorded in these Elbrus ice records over the time period of 1774-2013 CE. The $Ca^{2+}$ record reveals quasi-decadal variability with a generally increasing trend. Using multiple regression analysis, we found a statistically significant spatial correlation of the Elbrus $Ca^{2+}$ summer concentrations with precipitation and soil moisture content in the Levant region (specifically Syria and Iraq). The $Ca^{2+}$ record also correlates with drought indices in North Africa (r=0.67 p<0.001) and Middle East regions (r=0.71 p<0.001). Dust concentrations prominently increase in the ice core over the past 200 years confirming that the recent droughts in the Fertile Crescent (1998-2012 CE) present the most severe aridity experienced in at least the past two centuries. For the most recent 33 years recorded (1979-2012 CE), significant correlations exist between $Ca^{2+}$ and Pacific circulation indices (Pacific Decadal Oscillation, Southern Oscillation Index and Niño 4) which suggests that the increased frequency of extreme El Niño and La Niña events due to a warming climate has extended their influence to the Middle East. Evidence demonstrates that the increase of $Ca^{2+}$ concentration in the ice core cannot be attributed to human activities, such as coal combustion and cement production.

## 1 Introduction

Atmospheric dust is the most important aerosol emitted to the atmosphere in terms of mass (Knippertz & Stuut, 2014) and impacts. Despite the significance of atmospheric dust and its impacts on the planetary radiation balance, atmospheric chemistry, biosphere and human health (Middleton, 2017), knowledge of its regional variability and long term trends over past centuries is still poor (Mahowald et al., 2010). Dust concentration in the atmosphere depends on specific meteorological conditions, which may also be influenced by large scale circulation patterns (e.g. ENSO, NAO). Long term trends are

controlled by changes in precipitation and vegetation cover in dust source regions, with the vegetation cover being dependent on both natural (climatic changes) and anthropogenic (land cover change) causes. The complexity of dust emission, atmospheric transport and deposition mechanisms can result in large uncertainties in atmospheric dust models (Mahowald et al., 2007). The discrepancies between models are partly explained by limited observations of past dust variability, which has

limited possibilities for evaluating models' capability of reproducing of the dust cycle. However, reliable information on multiannual dust variability dating back to 1980 is now available from satellite data (e.g. Gautam et al., 2009; Chudnovsky et al., 2017; Li and Sokolik, 2018).

Analysis of recent aerosol patterns over different land and ocean regions show that despite significant trends over some major continental source regions, average values demonstrate little change in the past three decades (1980-2009) because opposite

trends cancel each other out in the global average (e.g. Chin et al., 2014). Recent broad-scale assessments of changes of dust emissions show a doubling of the dust deposition in many sedimentary achieves since the mid-18[th] century which was attributed to anthropogenic land use and short term climate variability (Hooper & Marx, 2018). Globally, anthropogenic dust sources account for 25% of emission, but this value varies and can be considerably higher–up to 75% in some regions (Ginoux et al., 2012). Climate-aerosol model simulations with the ability to separate natural and anthropogenic dust sources show that there

was a 25% increase in dust emissions between the 19[th] century and today. These changes are attributed to climate change (56%) and anthropogenic land cover change (40%) although the model underestimates dust concentrations in Asia, Middle East, and the US (Stanelle et al., 2014).

Records of past changes in dust concentrations are essential to better constrain interconnections between dust emissions and both natural and anthropogenic environmental changes. In this respect, proxy data are fundamental. Ice cores are natural

archives of past concentrations of various impurities present in the atmosphere, including dust (e.g. Legrand & Mayewski, 1997). Beyond dust, these records also grant insight into the strength of the particular dust sources and their changing magnitude through the time. Polar ice cores from Greenland and Antarctica reconstruct the changes of dust content in the atmosphere over hundreds thousands of years at a hemispherical scale (e.g., De Angelis et al., 1997; Delmonte et al., 2002; Legrand, 1987; Petit et al., 1999; Ruth et al., 2003). In contrast, data from ice cores drilled at mid-latitude mountain glaciers

proximal to arid areas reconstruct local- to regional-scale dust aerosol emission histories over shorter timescales (e.g. Grigholm et al., 2015, 2017; Kaspari et al., 2009; Osterberg et al., 2008; Thompson, 2000; Bohleber et al., 2018).

Mineral dust from North Africa and deserts of the Middle East is regularly deposited on glaciers in the Caucasus Mountains (Kutuzov et al., 2013). Due to its high elevation (over 5000 m a.s.l.) and proximity to arid and semi-arid areas, the Caucasus is a natural trap for desert dust. The absence of melt water infiltration near the summit of Elbrus ensures the preservation of a

climatic record in an ice core while high accumulation rates promote greater temporal resolution (Mikhalenko et al., 2015). A study focusing on the long-term trend of black carbon in the Elbrus ice (Lim et al., 2017) is currently underway as well as several studies investigating additional chemical species, including investigations of calcium and of sulfate. Data are discussed in two companion papers including the glaciochemistry of the deep Elbrus ice core drilled in 2009 (Preunkert et al., this issue).

Here, we report changes in $Ca^{2+}$ concentrations recorded in the Elbrus ice core between 1774-2012 CE and connections with natural variability, climatic and land use changes in the dust source regions.

## 2. Location, climatology, and backward air-mass trajectories

The Caucasus are situated between the Black and Caspian seas, and generally trend east–southeast, with the Greater Caucasus range often considered as the divide between Europe and Asia. The 2020 glaciers in the Caucasus cover an area of $1193 \pm 27$ km$^2$ (Tielidze & Wheate, 2018). Elbrus mountain glaciers contain about 10% of Caucasus ice volume and cover an area of 112.6 km$^2$ (Kutuzov, et al., 2015) (Fig. 1). Glaciers cover an altitudinal range from 2800 to 5642 m a.s.l. with the coldest conditions present above 5200 m a.s.l. where mean summer air temperature stays below 0°C.

To characterize possible sources of aerosols deposited on glaciers, we calculated three-dimensional backward trajectories of air parcels (elementary air particles) arriving at the Elbrus plateau (5100 m a.s.l.) using the NOAA HYSPLIT_4 trajectory model (Draxler & Hess, 1998; Stein et al., 2015) and NCEP/NCAR Reanalysis data on 2.5×2.5 degree grids (Kistler et al., 2001) for the 1948-2013 period. Ten day backward trajectories were calculated for every six hours for the whole period, resulting in a total of ~100k modelled backward trajectories. Over this time period, air particles transport is defined by the westerlies. Elbrus glaciers receive air particles which are very likely polluted with aerosols originating in the Mediterranean region: Turkey Eastern and Central Europe, the Middle East, North Africa and Southern Russia (Fig. 1b).

To identify potential dust source contributions, we also analysed vertical distribution of the backward trajectories. An objective criterion was chosen to extract locations of possible dust entrainment along the trajectories. The criterion is met when the air parcel is close to the ground (i.e. within the well-mixed boundary layer) allowing the uptake of mineral aerosols (Sodemann et al., 2006). Density plots were calculated only for ten day backward trajectories which descended below mixed layer depth. The depth is calculated by HYSPLIT_4 (using NCEP/NCAR Reanalysis data) for each point of backward trajectory as the height exceeding potential air temperature over surface air temperature by 2 K (Draxler & Hess, 1998).

## 3. Methodology

### 3.1. Ice core analysis

During August-September 2009, an ice core measuring 181.8 m in length was recovered at the western plateau of Elbrus in the central Caucasus (43°20′53.9′′ N, 42°25′36.0′′E; 5115 m asl). Drilling was performed in a dry borehole with a lightweight electromechanical drilling system. Borehole temperatures ranged from -17 °C at 10 m depth to -2.4 °C at 181 m (Mikhalenko et al., 2015). Ice cores were packed in insulated core boxes and shipped in a frozen state to the cold laboratory of the Institute des Géosciences de l'Environnement (IGE) in Grenoble, France for analyses. A total of 3724 samples were prepared from the surface to a depth of 168.6 m for the Elbrus core. Cores were subsampled and decontaminated at -15°C using the pre-cleaned electric plane tool methodology described by Preunkert & Legrand (2013). After the ice samples were cut using a band saw,

all surfaces of the ice samples were cleaned under a clean air bench by using a pre-cleaned electric plane tool over which the ice was slid. Sampling was continuous along the core. To control the decontamination efficiency process blank ice samples, consisting of ultrapure frozen MilliQ water were preceded regularly.

We determined cations ($Na^+$, $K^+$, $Mg^{2+}$, $Ca^{2+}$, and $NH_4^+$) using a Dionex ICS-1000 chromatograph equipped with a CS12
separator column. For anions, a Dionex 600 equipped with an AS11 separator column was used with an eluent mixture of $H_2O$, NaOH at 2.5 and 100 mM, and $CH_3OH$. A gradient pump system allows the determination of inorganic species ($Cl^-$, $NO_3^-$, and $SO_4^{2-}$) as well as short-chain carboxylates (Legrand et al., 2013). For all investigated ions, ice decontamination blanks were insignificant compared to respective concentrations in the ice cores.

### 3.2. Dating the ice core

Seasonal ice-core stratigraphy of chemical parameters and ice-core dating based on annual layer counting of the deep Elbrus ice core is described in detail in (Mikhalenko et al., 2015). The seasonal oscillations of $NH_4^+$ and succinic acid allow dating the core with seasonal resolution. Based on the ammonium and succinic acid profiles, each annual layer was divided into two parts corresponding to snow deposition under winter conditions and spring-summer-autumn conditions (Preunkert et al., 2000; Legrand et al., 2013; Mikhalenko et al., 2015). The annual counting was confirmed with one year uncertainty over the last one
hundred years by reference horizons from a 1963 tritium peak and the Katmai 1912 horizon (Mikhalenko et al., 2015). Though the annual counting becomes less straight forward prior to 1860, Mikhalenko et al. (2015) reported an ice age of 1825 at 156.6 m depth (i.e., 122.3 meter water equivalent, mwe). This time scale is consistent with the presence of a volcanic horizon at approximately 1833-1840 CE which can be attributed to the eruption of the Coseguina volcano in 1835. More recently, Preunkert et al. (this issue) extended the annual counting down to 168.5 m depth (i.e., 131.6 mwe) using the ammonium and
succinate records demonstrating that the seasonally-resolved record extends back to 1774.

Due to the glacier compression with depth, we applied a variable sampling resolution of 10 cm from 0 – 157 m, and then a sampling resolution of every 2 cm below 157 m depth. As a result (Fig. 3), the temporal resolution remains relatively consistent throughout the core, with 12 samples per summer over the 1950-2010 CE time period to 14 samples each summer over the 1900-1950 CE time period.

It should be noted that Fig. 2 shows the thickness of annual layers and does not represent the linear change in accumulation rate. In order to obtain an accumulation rate, layer thickness must be corrected for any compression which might have occurred following deposition (e.g. Paterson and Waddington, 1984) which is beyond the scope of this paper.

In addition, In June 2013 the existing ice record was expanded to include the years 2009 to 2012 CE via a 20.5 m long ice core which was extracted at the same drill site. In total, 515 samples of the firn core were analysed (85 samples per year).

For the 2013 ice core the succinic acid data were not available. We therefore used a combination of $NH_4^+$, BC and $\delta^{18}O$ profiles for annual counting and seasonal dissection. These results suggest that the 2013 firn core extends back to 2007, as confirmed by the similar patterns of $NH_4^+$, $Ca^{2+}$ and $\delta^{18}O$ between the first 3.8 mwe of the 2009 core and the 7.2-10.7 mwe interval of the 2013 core (Lim et al., 2017).

### 3.3. Climatic data analysis

In order to assess which factors drive the variations in dust content in the Elbrus ice core, we compared the dust concentrations with various climatic parameters in the potential dust sources. For climate data which may potentially influence the dust emission (temperature, precipitation, wind speed, soil moisture) we used ERA-Interim Reanalysis fields for the period of 1979-2013 CE (Dee et al., 2011). Since the variables used in this study are continuous and normally distributed the Pearson's correlation coefficient was used. The statistical significance of the correlation was checked using Student's t-distribution for probability. The correlations between $Ca^{2+}$ concentrations in the ice core and each climatological parameter fields were computed.

The Standardised Precipitation-Evapotranspiration Index (SPEI) was selected for use as a drought proxy (Vicente-Serrano et al., 2010) as its capacity for investigating of the relationship between dust emission and drought has been well documented (Achakulwisut et al., 2018). We considered SPEI calculated over periods of one, two, three, six and twelve months. The time series of SPEI were obtained by averaging over the regions of interest.

We also investigated the $Ca^{2+}$ record in relation with various circulation indices in order to identify the possible factors which influence the dryness in the dust sources and dust transport to the Caucasus Mountains. The following indices were analyzed: North Atlantic Oscillation (NAO), Pacific Decadal Oscillation (PDO), Atlantic Multidecadal Oscillation (AMO), Tripole Index for the Interdecadal Pacific Oscillation TPI (IPO) and El Niño-Southern Oscillation (ENSO) indexes, Southern Oscillation Index (SOI), Niño 3, Niño 3.4, Niño 4, Niño 1-2 (https://www.esrl.noaa.gov/psd/gcos_wgsp/Timeseries/). Correlations were calculated for periods of 30 years, using five year sliding windows. All series were detrended prior to analysis.

### 4. Results and discussion

The amount of dust in ice cores depends on many factors and corresponds to the presence of dust particles in the atmosphere. Primarily, dust emissions are influenced by the characteristics of the dust source (soil type, geomorphology, soil moisture) as well as by meteorological conditions (surface winds). Once dust clouds are uplifted to the mid troposphere, their transport depends on the main circulation patterns. In mountainous areas with high snow accumulation, wet deposition defines annual and seasonal aerosol concentrations. During spring and summer, the majority of air masses arrive to the Elbrus site from arid areas with calcareous soils. If all of the $Ca^{2+}$ in the Elbrus ice core is assumed to be of natural origin, then $Ca^{2+}$ concentrations can be considered a proxy for dust.

The total dust deposited at Mt. Elbrus may have three different natural components: (i) dust from local sources (nunataks, rock outcrops), (ii) sporadic arrival of large aeolian desert dust events and (iii) large scale background continuous terrigenous aerosol emissions. Apart from regions strongly impacted by sea-salt aerosols, the presence of calcium in aerosols in continental atmospheres is expected to mainly originate from exposed continental sediments. Even in polar regions (Antarctica or Greenland), calcium in ice mainly comes from dust where only a small fraction is related to sea-salt emitted from the ocean (De Angelis et al., 1997; Legrand, 1987). Assuming that $Na^+$ present in the Elbrus melted ice samples is only related to sea-

salt emissions and considering the $[Na^+]/[Ca^{2+}]$ ratio of 0.038 in seawater, we conclude that only $1.0\pm 0.7$ % in summer ($1.40 \pm 1.0$ % in winter) of total $Ca^{2+}$ may be attributable to sea-salt emissions. That percentage is clearly an upper limit since– in precipitation deposited at continental free tropospheric sites (e.g. Legrand, 2002)– $Na^+$ is not only related to sea-salt due to the presence of leachable sodium in alumino-silicate particles but also $Na^+$ from halide evaporates present in the deserts.

The volcanic rocks of Elbrus near the drilling site do not contain calcite; therefore, we can assume that $Ca^{2+}$ present in the Elbrus ice archives information on past dust emissions, including continuous background emissions and large dust plumes reaching the site. In the following sections, we attempt to separate these two possible calcium sources.

### 4.1. Calcium concentrations during desert dust events and background conditions

The majority of the calculated backward trajectories show a south-west origin with the highest frequency over the Middle East,
eastern Mediterranean and North Africa in all seasons. In winter (December-February), air masses tend to come from more remote locations whereas summer (June-August) reveals possible transport from the Caspian Sea region and Southern Russia (Fig. 3).

Large dust plumes reach the Caucasus Mountains which originate from the Middle East– and less frequently from the Sahara(Kutuzov et al., 2013). As seen in the Alps, these events impacts the chemistry of snow deposits to create calcium-rich
alkaline snow layers (Wagenbach et al., 1996). Deposition of these plumes increases the concentrations of numerous chemical species in Alpine ice due to either their presence in the dust at its emission stage or, being acidic, their interaction with alkaline material during transport (Usher et al., 2003). Deposition of light-absorbing impurities (in particular, black carbon and dust) plays an important role in changes of the snow and glaciers and enhances the response of the mountain cryosphere to climate changes via snow-albedo feedbacks (Ginot et al., 2014; Gabbi et al., 2015; Di Mauro et al., 2017; Skiles et al., 2018).

In this work, as well as in Preunkert et al. (this issue), Elbrus samples are considered to be impacted by dust events if two criteria are met: (1) they contain more than 120 ppb of calcium and (2) they fall below the 25% quartile of a robust spline calculated through the raw acidity profile. These selection criteria result in 616 dust-deposition summer samples (from a total of 2524) and 67 winter samples (from a total of 1150). Similar results were obtained when changing the calcium concentration criteria from 120 ppb to 100 or 140 ppb.

Using ammonium and succinate stratigraphy to separate the winter and summer seasons (Sect. 3.2), we determined half-year summer and winter means from 1774 to 2010 CE (Fig. 4). In Fig. 5, we report the seasonal cycle of calcium and ammonium averaged over two different periods of the 20th century (1900-1950 CE and 1950-2010 CE).

The mean concentration of $Ca^{2+}$ was 145 ppb with a maximum of 5506 ppb. Most of the dust from long distance transport is deposited on Elbrus during warm seasons. The average background summer $Ca^{2+}$ concentration was 103 ppb compared to 44
ppb in winter layers. Most of the large dust deposition events also occur during the warm season which further increases the difference between summer (178 ppb) and winter (54 ppb) layers (Table 1).

As seen in Fig. 6, most of long-term calcium trend is detected in summer with: (1) more frequent arrivals of large dust events after 1950 and (2) an increase of 100 ppb of the calcium background level after 1950. The maximum annual concentrations were found in 1999 and 2000 annual layers (980 and 850 ppb). There was a pronounced period of increased dust content in the 1960s with a following decrease in late 1970s. When compared to the background, exceptional peaks occurred during the following years: 1802, 1957, 1863, 1917, 1933, 1963, 1984, 1989, 1999, 2000, 2008 and 2009.

## 4.2. Increasing frequency of desert dust events

Warm season layers contain an increase in $Ca^{2+}$ throughout the duration of the investigated time period. The most pronounced increase in peak $Ca^{2+}$ concentrations occurred after the 1950s (Fig. 6). It is important to emphasize that this appearance of more frequent calcium peaks after 1950 (compared to the preceding decades) cannot be attributed to a decreasing time resolution of the record (e.g. via smoothing of events with depth) (Fig. 2). More specifically, the enhanced occurrence of summer calcium peaks between 1960 and 1970 compared 1950-1960 CE does not correspond to a significant decrease of the temporal resolution (from 9 samples over the 1960-1970 years compared to 11-12 samples over the 1950-1960 years).

In recent decades (1950-2010 CE), over which time period more dust events were detected in the Elbrus ice, a clear spring maximum of the dust fraction is observed (Fig. 5). These dust peaks are consistent with the timing of the arrival of large dust plumes at the site (Kutuzov et al., 2013). A major long-range dust outbreak event recently occurred over North Africa, as well as the eastern Mediterranean and Caucasus on 22 and 23 March 2018 (Solomos et al., 2018) which resulted in significant dust deposition on glaciers.

The calcium peaks in Elbrus ice containing dust are accompanied by an increase of magnesium (Fig. 7). Interestingly, the mean $[Mg^{2+}]/[Ca^{2+}]$ ratio in Elbrus samples containing dust (0.035, Fig. 7) is similar to those in atmospheric dust aerosols from the Sahara or Middle East. For instance, Koçak et al. (2012) reported dust event-related aerosol concentrations of sodium, magnesium and calcium from two Eastern Mediterranean sites: in Erdemli, Turkey with dust arriving from Middle East and from Heraklion, Crete with dust from Sahara. Importantly, the atmospheric concentrations of these cations also correspond to their water-soluble fraction (not the total fraction)– which were measured with Ion Chromatography– much like in the Elbrus ice core. In the case of Erdemli, during Middle East dust events, Koçak et al. (2012) reported atmospheric concentrations of 7085 ng m$^{-3}$ for $Ca^{2+}$ and 423 ng m$^{-3}$ for $Mg^{2+}$ (Table 2). Since Erdemli is located at 22 m above sea level and is situated 10 m away from the sea, a fraction of magnesium originates from sea-salt in addition to the leachable fraction of magnesium from dust. To correct concentration from the sea-salt contribution, we have used the $Na^+$ concentration (1148 ng m$^{-3}$) and assumed a mean $[Na^+]/[Ca^{2+}]$ ratio in dust of 0.08 as observed in Elbrus ice samples containing dust (not shown). Thus, neglecting the sea-salt calcium contribution, we estimate a dust sodium contribution of 567 ng m-3 (0.08 x 7085 ng m$^{-3}$). With that, and using the $[Mg^{2+}]/[Na^+]$ ratio in seawater (0.12), we estimate that 70 ng m$^{-3}$ of magnesium are originated from sea-salt and calculate a $[Mg^{2+}]/[Ca^{2+}]$ ratio for dust aerosol close to 0.05 (Table 2). A similar value is obtained for aerosol at Heraklion during a Saharan dust event (0.043, Table 2). The content of $Mg^{2+}$ in Elbrus samples impacted by dust is therefore very consistent with what is observed in atmospheric aerosol from the Eastern Mediterranean region during dust events.

## 4.3. Enhanced background concentrations

Figure 6 demonstrates an increase in the calcium background concentrations after 1950. This increase may be influenced by human activities, such as coal combustion and cement production, thereby contributing to the background calcium trend detected in the Elbrus ice. Similar impacts of anthropogenic emissions on various species in natural dust emissions (including calcium) were reported by Kalderon-Asael and colleagues (2009); this demonstrates that under strong stratification in the lower atmosphere in Israel, part of the atmospheric calcium may be anthropogenic nature. At the scale of Europe, Lee and Pacyna (1999) estimated that 0.8 Tg of anthropogenic calcium are emitted per year, coal combustion contributing for 60% and cement for 30% of total. However, to date, these anthropogenic calcium emissions remain one order of magnitude weaker than dust calcium emissions from North-East Africa (12 Tg yr$^{-1}$) or West Asia (12.7 Tg yr$^{-1}$) (Zhang et al., 2015).

Particles emitted during both coal combustion and cement production are rich in calcium (calcite). Therefore, we may expect a weaker $[Mg^{2+}]/[Ca^{2+}]$ ratio in particles emitted by these anthropogenic processes compared to that in natural dust particles. Examination of $Mg^{2+}$ and $Ca^{2+}$ in Elbrus ice layers that not impacted by dust events helps to determine a possible contribution of anthropogenic activities to the increasing background calcium trend in Elbrus ice. As seen in Fig. 8, the $[Mg^{2+}_{red.}]/[Ca^{2+}_{red.}]$ ratio in Elbrus summer ice is of 0.069 over the 1960-2010 compared to 0.126 over the 1774-1920. If we attribute this recent decrease of $[Mg^{2+}_{red.}]/[Ca^{2+}_{red.}]$ to a growing impact over the recent decades of calcium from cement (neglecting the presence of soluble magnesium in cement), the mean increase of $Mg^{2+}_{red.}$ level (10.8 ppb over the 1774-1920 years and 17.6 ppb over the 1960-2010 years) would lead to a $Ca^{2+}_{red.}$ increase of 55 ppb from 1774-1920 to 1960-2010. This value is half of the observed $Ca^{2+}_{red.}$ increase between the two time periods (71 ppb over 1774-1920 and 183 ppb over 1960-2010). These calculations suggest that half of the observed increase of background calcium concentration after 1950 are attributable to anthropogenic activities. As seen in Fig. 9, even prior to 1950 the $[Mg^{2+}_{red.}]/[Ca^{2+}_{red.}]$ ratio sometimes dropped to values as low as 0.1 or less (around 1820, 1850, 1865, or 1908). Although the time period 1940-1950 has a low ratio (0.09) the calcium concentrations moderately increased (27 ppb; less than a third of the overall increase after 1950). Figure 9 suggests that a decrease in the $[Mg^{2+}_{red.}]/[Ca^{2+}_{red.}]$ ratio is not always associated with enhanced calcium levels but that enhanced magnesium levels also lead to an enhancement of the $[Mg^{2+}_{red.}]/[Ca^{2+}_{red.}]$ ratio. For instance, over 1853-1861, a mean $[Mg^{2+}_{red.}]/[Ca^{2+}_{red.}]$ ratio of 0.22 occurs at the same time as a relatively high $Mg^{2+}_{red.}$ level (16 ppb instead of 10.8 ppb over the 1774-1920 time period).

De Angelis & Gaudichet (1991) presented additional evidence to suggest that cement production and use (despite its growing impact) does not represent the dominate contribution to background calcium levels at Elbrus derives in their study of calcium and aluminium at the Col du Dome site, Mont Blanc, France. The authors demonstrate that the increase of both dust arrival frequency in the 1980s and the background dust levels occurred without any coinciding decrease in the aluminium to calcium ratio. Given the low aluminium content of cement as compared to desert dust, these observations suggest that the Col du Dome site is not significantly impacted by the growing use of cement. We may expect that this impact is even weaker at the Elbrus site.

## 4.4. Climatic factors

Two major dust sources contribute mineral particles to the Elbrus glaciers: the Sahara and the Middle East. It was established that majority of the small scale dust sources in the Middle East are located in northern Mesopotamia (northern Syria – north-western Iraq) and the Syrian Desert (Kutuzov et al., 2013). The Levant is a major source of atmospheric dust (Middleton, 1986) with natural, anthropogenic and hydrological (intermittent streams and lakes) sources. The area between the Tigris and the Euphrates in Iraq contains natural desert dust sources while the Neinava region in Iraq was recently identified as the most active dust source in the Middle East (Moridnejad et al., 2015). In the northern Sahara, strong sporadic dust storms originating in the Libyan desert and the foothills of the Ahaggar Mountains in eastern Algeria sometimes reach the Caucasus in the spring (Kutuzov et al., 2013).

A statistically significant spatial correlation occurs between $Ca^{2+}$ and precipitation as well as between soil moisture content in the Middle East (Fig. 10). Unlike in the Middle East, no spatial correlation was found between $Ca^{2+}$ and the amount of precipitation in North Africa. However, it should be noted that the observation network is very sparse in the arid areas and there are much larger uncertainties in Global Precipitation Climatology Centre (GPCC) datasets in North Africa.

The $Ca^{2+}$ record was also compared with drought indices in potential dust sources in the Middle East (32-37° N; 38-45° E) and in North Africa (20-35° N; 0-35° E) (Fig. 10). $Ca^{2+}$ significantly correlates with drought indices for both regions. The highest correlations were found for the SPEI 3 index which is determined by aridity over the three previous months. The correlation coefficient for the Middle East is statistically significant (p<0.001) and reaches -0.71. The correlation remains significant after trend removal (r=-0.48, p<0.001 for 1904-2012 CE and -0.63 (p<0.001) for 1970-2012) (Fig. 11b, d). Periods of dryer conditions in the Middle East region coincide with the increased $Ca^{2+}$ concentrations and vice versa. During the period of droughts in both regions, a greater amount of mineral particles is emitted into the atmosphere and transported to Caucasus glaciers during the spring and summer. The general increasing trend in dust content corresponds to negative trends in precipitation and increased dryness of the soil.

$Ca^{2+}$ correlates significantly with the SPEI 3 drought index for the North African region (r=0.67 p<0.001), but most of the correlation is defined by similar tendencies in two series. Drier conditions at the dust source region correspond to a positive trend in dust concentrations in the Elbrus ice core. The correlation weakens for the full detrended series (r=0.27, p<0.05) and increases in the later period, reaching 0.73 (p<0.001) in 1970-2012 CE (Fig. 11a, c). Large portions of the annual dust flux in Elbrus can be deposited during a single strong deposition event. Such events are more often associated with dust transport from the Sahara (Kutuzov et al., 2013). Dust emitted from North Africa can also mix with dust from Middle Eastern sources during transport (Shahgedanova et al., 2013).

As evidenced by the Elbrus ice core record, the frequency of dust events and total $Ca^{2+}$ concentrations has increased, and the most notable trend occurs after the 1950s. This increase corresponds with the recent analysis of centennial climate change trends in Africa which depict a significant northward expansion of the Saharan desert in the winter (Thomas & Nigam, 2018). The desert conditions are observed across larger territories in recent years and are associated with a slightly negative trend in

winter precipitation and an increase in surface air temperatures. In coastal North Africa, dry conditions dominated and rainfall decreased since the 1980s (Nicholson et al., 2018). The number of dust days in the eastern Mediterranean increased over the period 1958–2006 CE (Ganor et al., 2010).

Two periods of maximum $Ca^{2+}$ concentrations in the ice core correspond to the two most severe drought episodes in the Middle East since the1940s, and occur during 1998–2000 CE and 2007-2009 CE (Barlow et al., 2016). The significant precipitation decline (up to 70%) during these years is explained by the dominance of high pressure systems over the eastern Mediterranean during the winter and spring months (Trigo et al., 2010).

Our findings are supported through analysis of the frequency of droughts in Syria. For the period between 1961 to 2009 CE, droughts were observed in 25 of the examined years, resulting in ~40% of the years classified as drought years. On average, droughts lasted 4.5 years, although in the 1970s a single drought lasted ten years (Breisinger et al., 2011). Droughts lasting two or more years significantly impacted agricultural production and livestock in the north-east of the country, where a single drought in 1961 resulted in the loss of 80% of camels and 50% of sheep. During the drought of 1998–2001 CE, 329,000 people of which 47,000 were nomadic families were forced to eliminate livestock numbers and experienced an acute shortage of food (De Châtel, 2014).

Anthropogenic land use and changes in land cover impact the soil erodibility and dust emission. The magnitude of such impacts is highly uncertain as both climatic and anthropogenic processes occur simultaneously (Webb & Pierre, 2018). Unsustainable agricultural practices, overgrazing, and deforestation may significantly increase the area of the dust sources. It should be noted that only around 5% of the land in North Africa and the Middle East is suitable for agriculture; the rest consists of pastures, forests, shrubs, urban zones, badlands, rocky areas, and deserts (Sivakumar & Stefanski, 2007).

### 4.4.1. Atmospheric circulation patterns

The general increase in dust concentrations in the Elbrus ice core are accompanied by a quasi-decadal variability. The relationship of precipitation in the Middle East region to the different large-scale circulation patterns are summarized in a recent review of the droughts in the Middle East and Southwest Asia (Barlow et al., 2016 and references therein). The precipitation and occurrence of droughts in the Middle East region may be influenced by several major climatic features such as the NAO, southwest Asian and Indian monsoon, global-scale variability associated with ENSO and state of the western Pacific which together determine the strength of the general atmospheric circulation (Barlow et al., 2016).

For the 33 year time period (1979-2012), significant correlations exist between $Ca^{2+}$ and PDO, SOI and Niño 4 in the preceding winter period (Table 3). A significant correlation (r=0.47 p<0.01) for the Niño 4 index also exists for Dec-June in 1948-2012. The relationship between dust concentrations and circulation indices of the Pacific Ocean is illustrated by the spatial correlation of $Ca^{2+}$ with sea surface temperatures (Supplementary Fig. 1). 23 out of 47 years of high dust deposition recorded in Elbrus ice core coincide with negative Niño 4 phases in 1900-2012 CE. After the 1950s, 19 out of 23 years of increased dust concentration correspond to the negative Niño 4 and negative PDO (Fig. 12). No statistically significant correlation was found between the dust in the Elbrus ice core and the NAO index. $Ca^{2+}$ also correlates with the SOI index of the preceding autumn

and winter (Nov-Jan) over the last 50 years. It is still unclear whether circulation in the Pacific played a significant role in the precipitation patterns in these dust source regions before the mid-20th century or not. The uncertainties in reanalysis data in this period are large and therefore do not allow us to draw any conclusions about the stability of the revealed correlations in time.

A significant negative correlation was found between atmospheric dust concentrations in Syria and the PDO in springtime during 2003–2015 CE. It was shown that a positive geopotential height anomaly is formed over the Arabian Peninsula and North Africa during the positive phases of PDO (Pu & Ginoux, 2016). A positive PDO is characterised by an increase of cyclonic activity over the northern Pacific and northern Atlantic which occurs together with the intensification of subtropical antyciclones. Despite an increase in geopotential height, the positive PDO increases the probability of moisture transport to

North Africa and Middle Eastern regions in the spring (Dai, 2013; Pu & Ginoux, 2016). This moisture is due to increased advection from the Mediterranean Sea which causes deep convection due to unstable stratification in the lower troposphere. The negative PDO phases on the other hand are associated with a negative geopotential height anomaly in the middle troposphere over the Mediterranean and the Middle East and cyclonic activity in the region. Low pressure anomalies over Europe, the Southern Arabian Peninsula and north-eastern to eastern Africa create favourable conditions for westerly winds

from North Africa and increase the probability of dust transport to the Caucasus.

Studies of interannual decadal variability of dust activity in the Arabian peninsula and Fertile Crescent suggest that the occurrence of severe droughts and increased dust emission were influenced by the La Niña phase amplified by the negative PDO (Notaro et al., 2015). There are still large uncertainties in such connections due the lack of long-term observations (Pu & Ginoux, 2016). The correlation between Pacific circulation indices and the amount of dust in the Elbrus ice core supports these

previous findings and may indicate an increase in the frequency of extreme El Niño and La Niña events with climate warming (Cai et al., 2015) which can then influence circulation over the Pacific Ocean and extend to the Middle East.

Our results are also in line with previous conclusions about the possible influence of large circulation patterns on the aridity of the tropics. Dai (2011) suggests that the El Niño-Southern Oscillation, tropical Atlantic sea surface temperature (SST), and Asian monsoons played a significant role in the increase in global aridity since the 1970s over Africa, southern Europe, East

and South Asia, and eastern Australia while recent warming has increased atmospheric moisture demand contributing to the drying. It is expected that due to anthropogenic climate change the tropical belt may expand toward the poles and shift precipitation patterns, which ultimately will lead to an increase in the territory affected by droughts (Seidel et al., 2008). Model studies show that under global warming the large-scale circulation systems (jet streams and storm tracks) may shift poleward (Mbengue & Schneider, 2017). The size and intensity of the Hadley cell and the associated shift of the subtropical anticyclone

zone are likely to occur over the 21st century, which should primarily affect the precipitation regime in subtropical latitudes (Lu et al., 2007). An expansion of the global tropics since 1979 by 1° to 3° latitude was identified in both hemispheres although the mechanisms behind this tropical expansion are still unclear (Lucas et al., 2014).

## 4.5. Comparison to other paleo records

The first record of dust deposition history in the Caucasus was obtained by F.F. Davitaya in 1962 (Davitaya, 1969). This work was based on sampling of the firn layers from a crevasse located at the Kazbek plateau (4600 m asl). Based on dust layer counting Davitaya estimated that sampled firn layers covered the period between 1793 and 1962. Dust concentrations clearly increased by a factor of 3 since the late 1920's. This increase was attributed to various reasons including local dust influence due to glacier retreat, industry development, fires, World War II and volcanic activity. Despite different methodology and location, we qualitatively determined a similar long-term dust trend in the Elbrus ice core record. The average $Ca^{2+}$ concentration over the same periods increases by a factor of 2.5 from 88 ppb in 1793-1925 CE to 224 ppb in 1925-1962 CE. An overall increasing trend in dust content and $Ca^{2+}$ exists in Colle Gnifetti (Alps) ice cores after 1870 (Thevenon et al., 2009). Unlike in the Elbrus $Ca^{2+}$ records, no significant long-term trend over the past 300 years exists in the Alps (Bohleber et al., 2018). In Central Asia, negative trends in dust concentrations were found in ice cores in Tien Shan (Grigholm et al., 2017) and in Tibetan Plateau (Grigholm et al., 2015). After the 1950s, calcium concentrations and multiannual and decadal variability significantly decrease in both Central Asian regions due to changes in dust storm frequency in arid areas related to regional trends in reduced zonal wind strengths (Grigholm et al., 2015).

A review of the dust paleo records demonstrates that 16 of the 25 compiled sedimentary archives from across the globe depict a doubling in dust emissions over the past ~250 years which was attributed to the impact of anthropogenic activity in source regions (Hooper and Max, 2018). Previous estimates for the atmospheric dust variability also suggests that desert dust doubled during the 20th century with some exceptions (Mahowald et al., 2010). All of the existing regional and global estimates of atmospheric dust variations in the past have large uncertainties due to limited and unevenly distributed paleodata records, which is especially true for the important dust sources of the North Africa, East Asia and Middle East/Central Asia (Mahowald et al., 2010). Tree ring reconstructions of the June-August drought variability over the past 900 years in the Mediterranean revealed that 1998-2012 CE was the driest period in the Levant since the 12th century (Cook et al., 2016). Similarly, the latter half of the 20th century was found to be one of the driest in the last nine centuries in north-western Africa (Touchan et al., 2011).

Information about variations in dust concentrations over the Caucasus leads to a better understanding of the climate change consequences and atmospheric circulation patterns in the dust source regions. Arid and semi-arid regions in North Africa and the Middle East are unstable under the recent climatic changes. Temperature and hydrological anomalies during the last millennium have led to large variations in human migration patterns and agricultural production, with precipitation variability as a key factor in the productivity in the Middle East region (Kaniewski, 2012). The Elbrus ice core record confirms previous findings (Cook et al., 2016; Kelley et al., 2015) that the recent droughts in 1998-2012 CE were the most severe over at least the past three centuries.

## 5. Conclusions

This paper presents the $Ca^{2+}$ record from the Elbrus ice core and provides a valuable archive of regional atmospheric dust variability in the past. $Ca^{2+}$ concentrations in the Elbrus ice core are a proxy for atmospheric dust transported over long distances from primary dust sources in North Africa and the Middle East. The $Ca^{2+}$ record shows that recent decades were characterized by the highest dust emission activity since 1774. From the 1970s onward, Saharan dust deposition events significantly impact the dust deposition in the Caucasus. The increase in frequency and intensity of dust deposition events is related to an overall trend towards dryer conditions and consequently an increased probability of dust emission from the aforementioned dust production sites. $Ca^{2+}$ significantly spatially correlates with precipitation and soil moisture content in the Middle East. The correlation between circulation indices and the amount of dust in the Elbrus ice core indicate the influence of circulation over the Pacific Ocean extended to the Middle East.

## Data availability

Calcium data can be made available for scientific purposes upon request to the authors (contact: kutuzov@igras.ru; suzanne.Preunkert@univ-grenoble-alpes.fr or michel.legrand@univ-grenoble-alpes).

## Author contributions

S. Kutuzov performed field research, analyzed data and wrote the original manuscript. M. Legrand analyzed ice samples, assessed data and contributed to the original manuscript. S. Preunkert analyzed ice samples and provided comments on the original manuscript. P. Ginot and V.Mikhalenko performed analysis and provided comments on the original manuscript. K. Shukurov calculated backward trajectories and provided comments on the original manuscript. A. Poliukhov and P. Toropov analyzed climatic data, assessed atmospheric circulation patterns and provided comments on the original manuscript.

## Acknowledgments

Research was supported by RSF grant 17-17-01270. The Les Enveloppes Fluides et l'Environnement- Chimie Atmosphérique (CNRS) program entitled "Evolution séculaire de la charge et composition de l'aérosol organique au dessus de l'Europe" provided funding for analysis in France, with the support of Agence de l'Environnement et de la Maîtrise de l'Energie. Backward trajectory analysis was supported by the Presidium Program of the Russian Academy of Sciences, No. 20. Interpretation of the shallow core was supported by the President Grants for Government Support of Young Russian Scientists and the Leading Scientific Schools of the Russian Federation grant No.MK-2508.2017.5. We thank Dr. N.M Kehrwald and F.S. Schoessow for valuable comments on the original manuscript and English language editing.

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

**Table 1.** $Ca^{2+}$ concentrations in Elbrus ice cores from different periods– including total concentrations and excluding background samples from the large dust deposition events.

| Period | $Ca^{2+}$ concentration (*ppb*) | | | |
|---|---|---|---|---|
| | summer (total) | summer (background) | winter (total) | winter (background) |
| 1774-1800 | 65 | 64 | 42 | 42 |
| 1800-1850 | 78 | 69 | 52 | 46 |
| 1850-1900 | 100 | 69 | 43 | 36 |
| 1900-1950 | 156 | 83 | 37 | 33 |
| 1950-2000 | 344 | 181 | 75 | 58 |
| 2000-2012 | 439 | 212 | 99 | 64 |
| 1774-2012 | 172 | 103 | 54 | 44 |

**Table 2.** Composition of aerosol collected at Erdemli during dust events from Middle East and at Heraklion during dust events from Sahara (from Koçak et al., 2012). The magnesium sea-salt contributions were calculated via the $Na^+$ content after subtracting its dust contribution that was calculated as 0.08 times the $Ca^{2+}$ content (see text).

| Species | Erdemli (October 2007) | Heraklion (April 2008) |
|---|---|---|
| $[Na^+]$ | 1148 ng m$^{-3}$ (sea-salt: 581 ng m$^{-3}$) | 1106 ng m$^{-3}$ (sea-salt: 445 ng m$^{-3}$) |
| $[Mg^{2+}]$ | 423 ng m$^{-3}$ (sea-salt: 70 ng m$^{-3}$) | 407 ng m$^{-3}$ (sea-salt: 53 ng m$^{-3}$) |
| $[Ca^{2+}]$ | 7085 ng m$^{-3}$ | 8264 ng m$^{-3}$ |
| $[Mg^{2+}]/[Ca^{2+}]$ in dust | 0.050 | 0.043 |

**Table 3.** Correlation coefficient of the Elbrus $Ca^{2+}$ data and large-scale atmospheric circulation indices.

| | 1948-2012 | 1979-2012 |
|---|---|---|
| PDO (Dec-June) | -0.35 | -0.42 |
| SOI (Oct-Jan) | 0.40 | 0.49 |
| Niño 4 (Dec-June) | -0.47 | -0.57 |

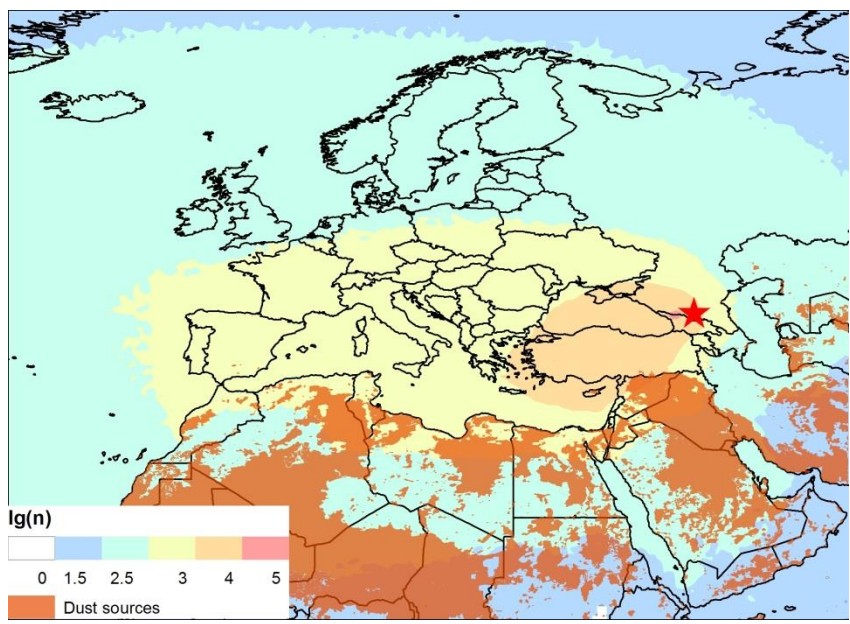

**Figure 1: Location of Elbrus (red star) and dust sources (orange polygons) based on (Ginoux et al., 2012). Annual NOAA HYSPLIT_4 10-day backward trajectory density plots for the period 1948-2013 using NCEP/NCAR Reanalysis. Trajectories were run every 6 hrs.**

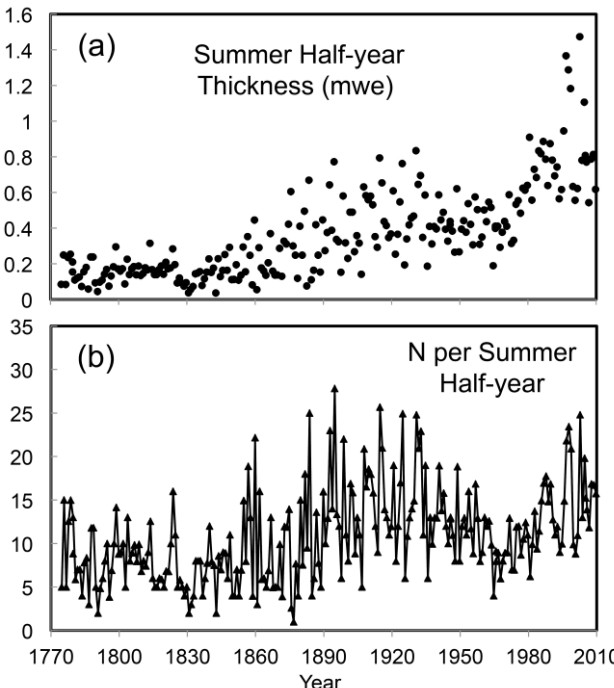

**Figure 2: (a): Mean summer half-year thickness along the Elbrus deep ice core. (b): Numbers of samples (N) spanning summer half-years. See details in Preunkert et al., this issue.**

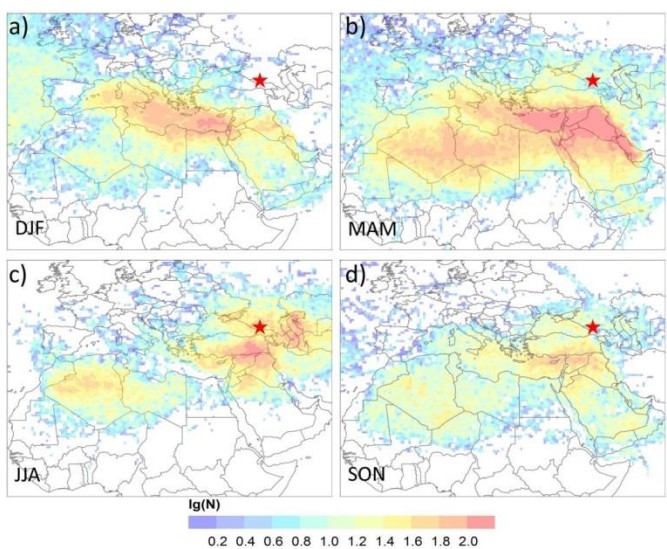

**Figure 3: NOAA HYSPLIT_4 10-day backward trajectories density plots for the period 1948-2013 for December – February (a), March-May (b), June-August (c) and September-November (d) using NCEP/NCAR Reanalysis. Trajectories were run every 6 hrs. Only backward trajectories within the boundary layer were considered.**

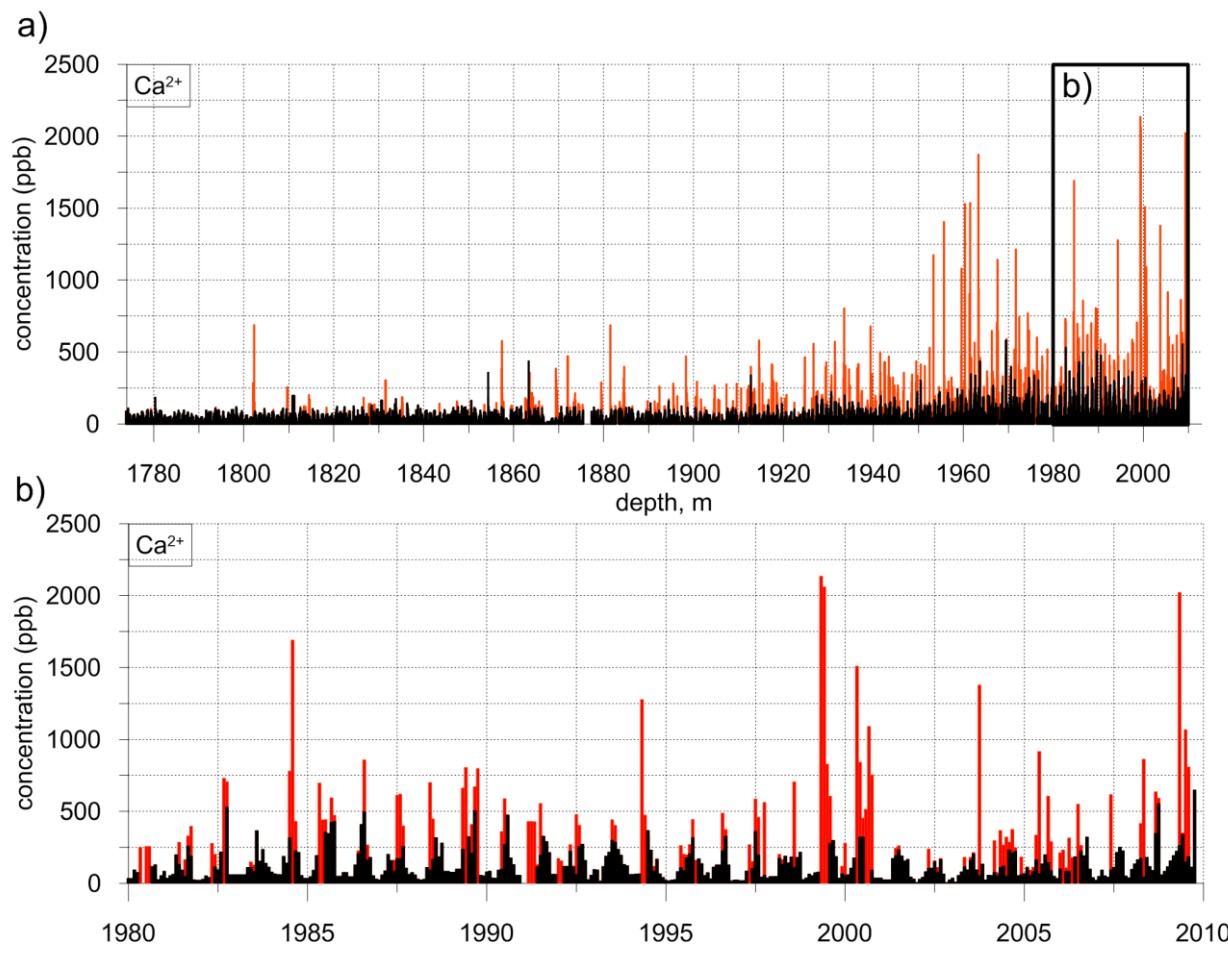

**Figure 4: (a): Total (red) and background (black) Ca²⁺ concentrations (ppb) in Elbrus ice core samples. The full record (a) and the upper section of the ice core (b) are shown. Note: background concentration values reflect removal of samples which were found to have been influenced by large dust deposition events.**

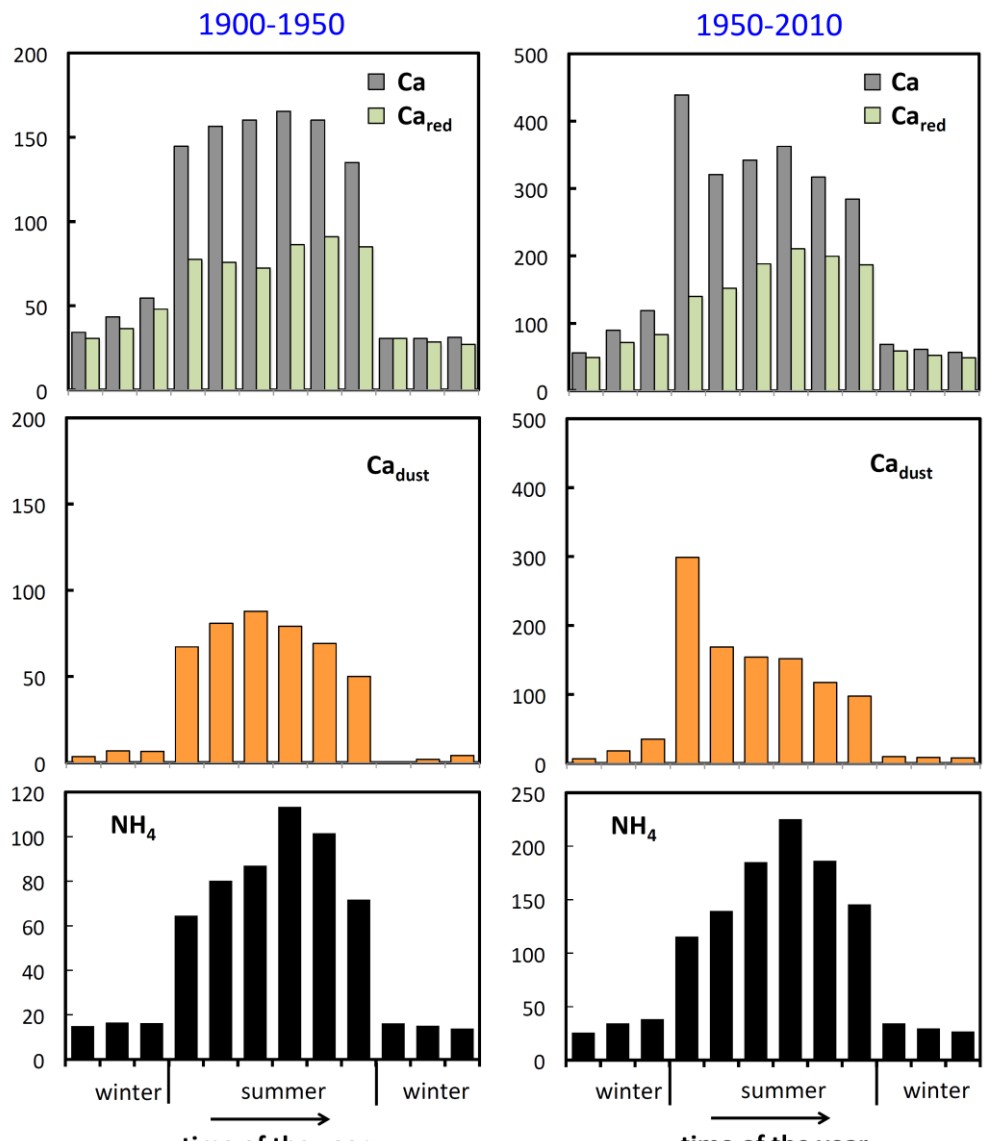

**Figure 5: (a): Averaged concentrations of calcium and ammonium (used for determining seasonality as described in the text), over 1900-1950 CE (left) and 1950-2010 CE (right). Total calcium concentrations are the grey bars in the top panels while calcium concentrations calculated after the removal of samples are in red. The middle panels refer to calcium concentrations corresponding to the dust fraction (i.e., $[Ca^{2+}]$ - $[Ca^{2+red}]$, orange bars). Note: The equidistant binning of the summer and winter layers applied in order to derive these 12 values per year does not necessary correspond to monthly means. This would only be the case if there were an absence of seasonality in precipitation. Nevertheless, the figure clearly illustrates the occurrence of a dust related calcium peak at the beginning of the summer season.**

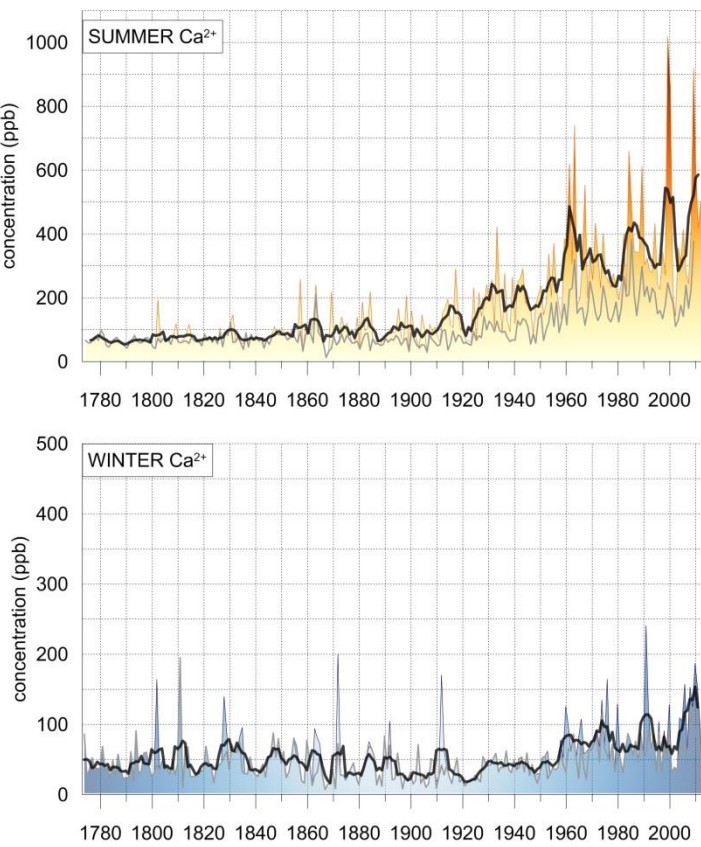

**Figure 6: Total and background (grey lines) Ca²⁺ concentrations in Elbrus ice core during summer and winter. Five-year moving averages are shown as black lines.**

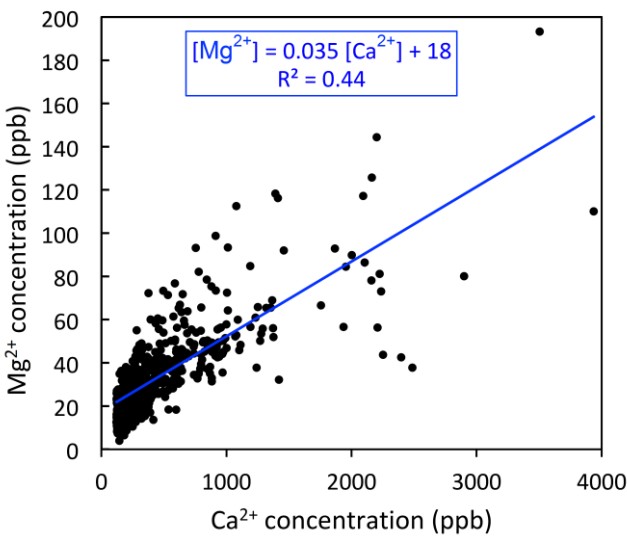

5  **Figure 7: Magnesium versus calcium concentrations (in ppb) in dust ice samples from Elbrus.**

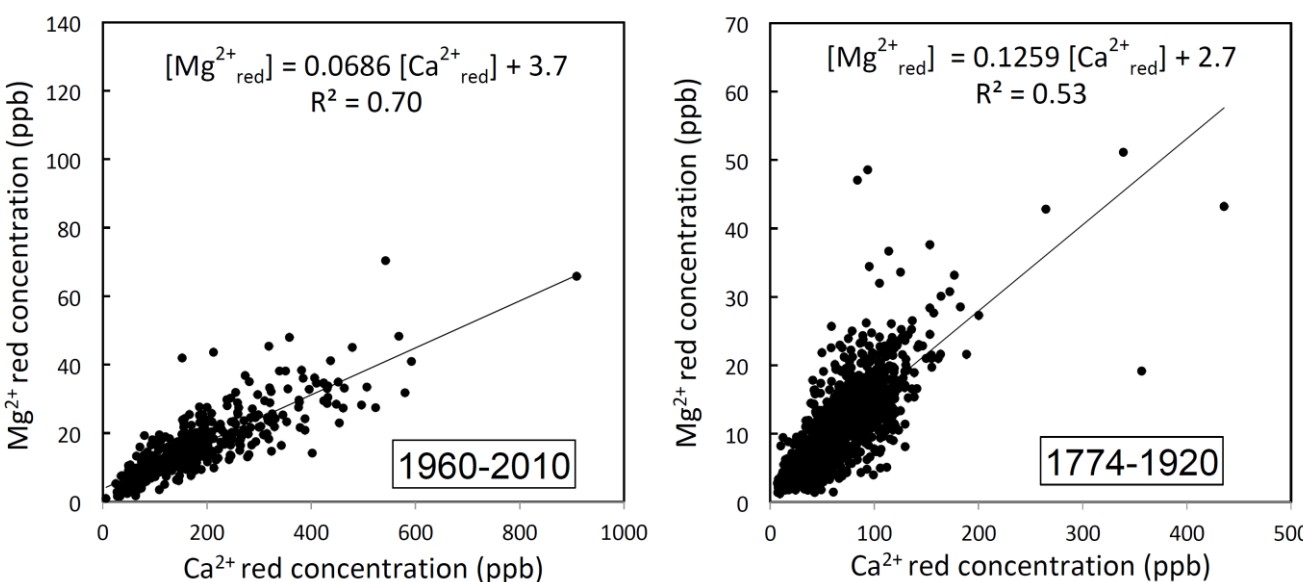

**Figure 8: [Mg²⁺red.] versus [Ca²⁺red.] in individual Elbrus ice summer samples (not impacted by dust events) over the recent decades (left) and prior to the 1950 increase in calcium.**

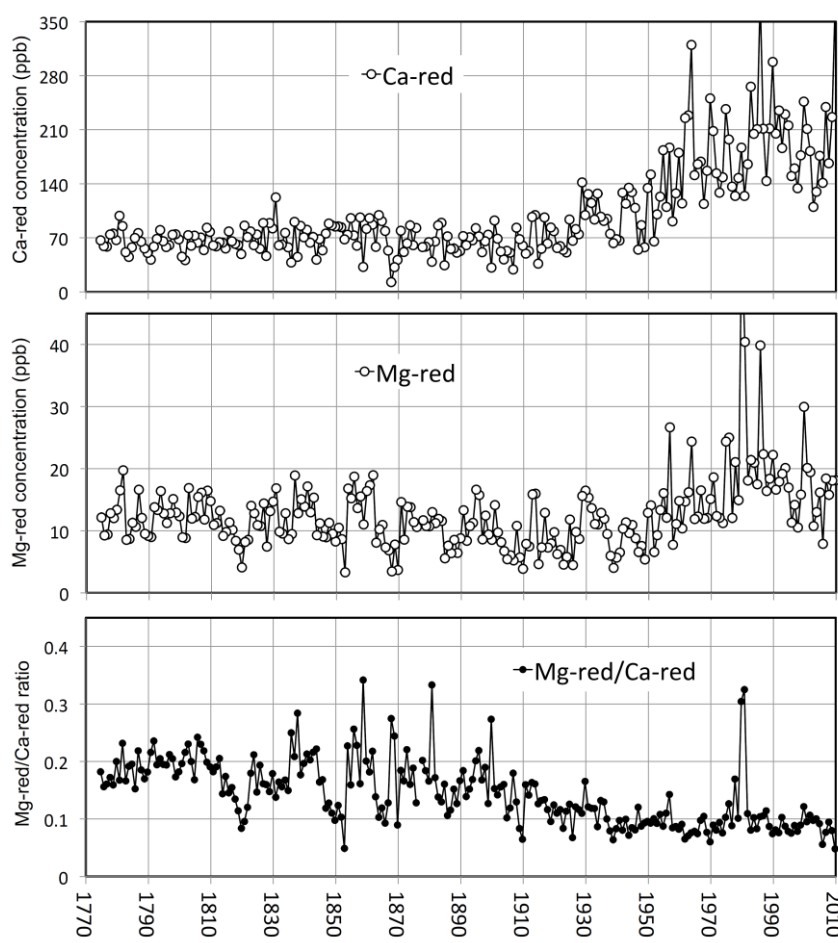

**Figure 9: Individual summer means of background levels of Ca$^{2+}$, Mg$^{2+}$, and of the [Mg$^{2+}_{red.}$]/[Ca$^{2+}_{red.}$] mass ratio along the Elbrus ice cores.**

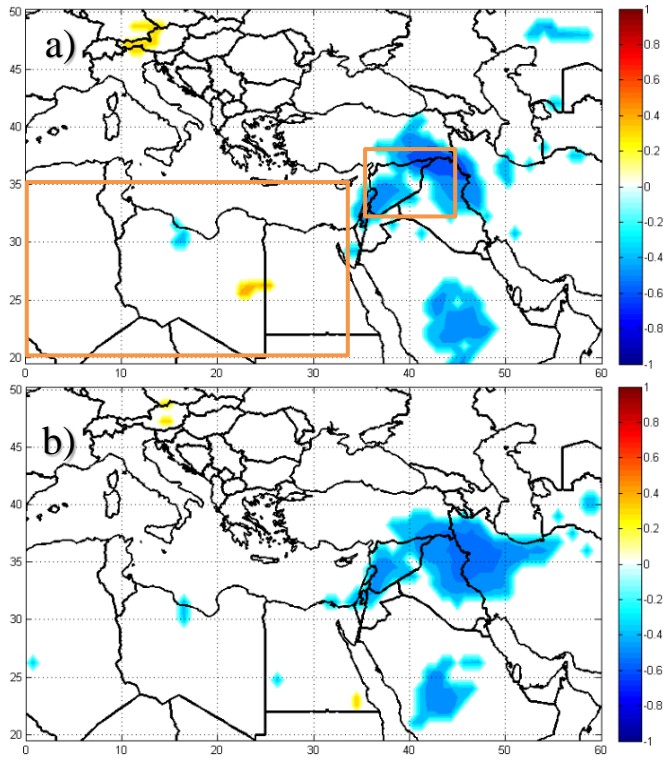

**Figure 10: Spatial correlation of Elbrus Ca²⁺ concentrations with soil moisture (a), and precipitation (b) ERA-Interim from 1979 to 2013. North African and Middle Eastern domains are shown by orange boxes.**

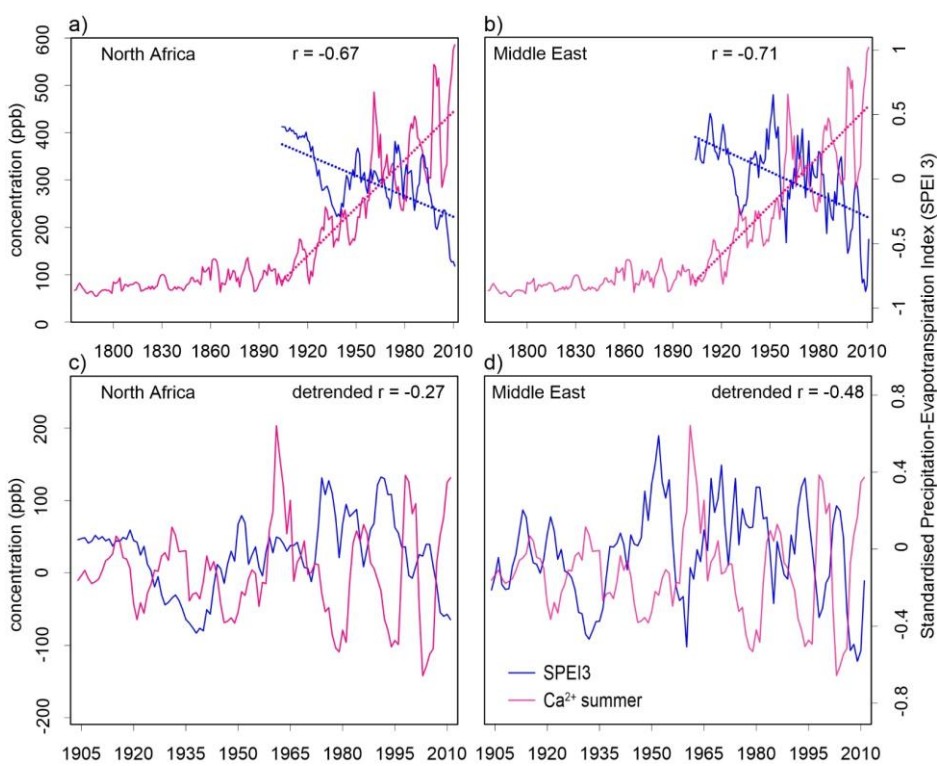

**Figure 11: Ca²⁺ and Standardised Precipitation-Evapotranspiration Index (SPEI3) averaged over the regions of (a), (c) North Africa (20-35°N; 0-35°E; and (b), (d) the Middle East (32-37°N; 38-45°E). Outlines of the regions are presented on Figure 10. Detrended records are shown at the bottom graphs.**

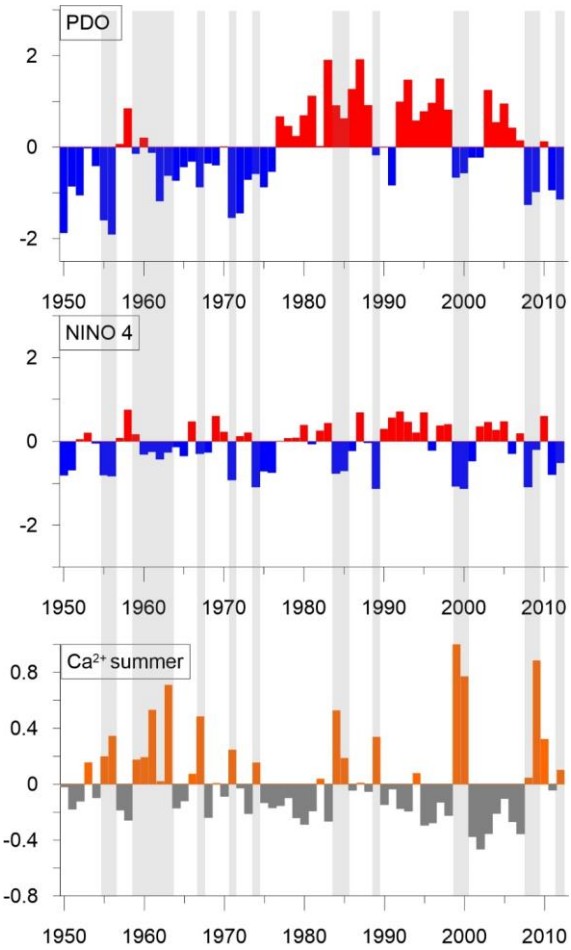

**Figure 12: PDO (a), Niño 4indices (December-June average) (b) and normalized Ca2+ record (c). Grey shading indicates years when high dust concentrations coincided with negative Niño 4 phases.**