# Peer review of "Figure S1: Correlation between Elbrus $\text{Ca}^{2+}$ concentrations and the December-March sea-surface temperature over the 1979-2012 period."

_Atmospheric Chemistry and Physics, 2019_

## Referee Comment (RC1) · Anonymous Referee #1 · 5 Jun 2019

The manuscript by Kutuzov et al. presents a calcium-dust proxy record from an ice core drilled on Mt. Elbrus, Caucasus, spanning the time period 1774-2013 CE. The discussion on the dust proxy include separating the background signal and the main dust events, evaluating the frequency and amplitude of dust events, establishing a relation with the potential dust sources by means of analysis of atmospheric circulation patterns and climate indices. I found the dataset very interesting per se, which warrants publication in ACP. The discussion and interpretation of the record is quite detailed and includes very interesting findings, following an approach established by the same group of authors. However, I also found that one relevant issue, related to the almost 10-fold increase in snow accumulation with potential implications on the interpretation of the proxy record, is not discussed. Therefore I recommend a major review of the

manuscript.

General comments

After reading the manuscript, I am not sure whether the trends mostly reflect changes in deposition/accumulation processes rather than changes in dust emissions. This aspect is not discussed in the manuscript, while I believe it is central for the interpretation of the record. I elaborate on this consideration in the lines below.

In Figure 1a one can be appreciate how the summer half-year thickness, expressed in meters of water equivalent, increases significantly since the beginning of the record. This implies increased snow accumulation, in addition to the expected ice compaction with depth. In fact the "companion" discussion paper by Preunkert et al., reports a "decrease of the net annual snow accumulation from 1.5 mwe (0.8 mwe in summer and 0.7 mwe in winter) near the surface to 0.18 mwe (0.15 mwe in summer and 0.03 mwe in winter) at 157 m depth". Therefore we see an almost 10-fold increase in snow accumulation rates along the core. The authors discussed the related potential issues in determining the the ability to detect the frequency of dust events; in order to overcome this issue, they adopted a strategy with finer sampling in the bottom sections of the core. While this precaution is an effective measure to that aim, it does not respond to the issue of whether the increased accumulation rates reflect increased precipitation and wet scavenging, in other words a larger or more frequent sampling of the atmospheric dust loading during precipitation events. As a result, it cannot be safely concluded which effect primarily (or maybe both) determines the observed trends in the dust proxy. This kind of reasoning is partly grounded in the long-standing debate on whether for instance dust concentrations or deposition fluxes are a better proxy for atmospheric dust / dust variations (e.g. Fischer et al., 2007; Mahowald et al., 2011).

I recommend that these issues are throughly discussed in the manuscript, and the interpretations and conclusions weighted accordingly.

Specific comments

p. 2 / lines 3-4: Please explain how

2/4-5: please provide some references

2/8: it would seem more precise to say that many of the archives reported in the cited manuscript show a doubling of the respective dust signals

2/9-13: Given the level of detail reported here with reference to the cited paper, it may be worth reporting other studies as well (e.g. Ginoux et al., 2012; older papers assessing the issue at the global level)

2/25: Two references are listed as Kutuzov et al. 2015 a,b. Please delete the non-relevant one. In addition, remove from the reference list the discussion paper (Kutuzov et al., 2013).

3/9: Make sure that the special character is properly displayed. In addition, in the legends of Figures 1 and 2, a resolution of 0.5 x 0.5 degree is reported. Which one is correct?

3/12: Rather than aerosols, the HYSPLIT analysis reported here shows that "Elbrus glaciers receive AIR MASSES from sources . . ."

3/14-21: Please clarify whether the density plots in Figure 2 are based solely on the the back-trajectories passing close to the ground (and what about Figure 1?). In addition, please explain how did you define the well-mixed boundary layer.

3/25: Please report the geographical coordinates

3/26: ranged from . . . AT 10 m depth . . .

3/30: Could you report in a few words the main aspects of that methodology?

3/30-31: Please specify whether the sampling is continuous along the core

4/8-9: Define what is meant by decontamination blank

4/18-19: Please rephrase

5/7-8: Please report briefly the methodology of the cited paper, i.e. how one can identify dust events on the basis of Ca2+ and acidity records.

5/20-22: It would be interesting to estimate/report the uncertainty arising from this assumption, and propagate it to the dust proxy.

5/22-25: Can you provide an estimate of how much this uncertainty could amount to?

5/31 - 6/3: It is not clear what you mean by "disturb" in both sentences. Please rephrase.

6/8: What do yo mean by "warm periods"? Warm years/decades? Warm seasons?

References

Fischer, H., Siggaard-Andersen, M. L., Ruth, U., Rothlisberger, R., and Wolff, E. W.: Glacial/Interglacial changes in mineral dust and sea-salt records in polar ice cores: sources, transport, and deposition, Rev. Geophys., 45, RG1002, doi:10.1029/2005RG000192, 2007.

Ginoux, P., Prospero, J., Gill, T., Hsu, N., and Zhao, M.: Global-scale attribution of anthropogenic and natural dust sources and their emission rates based on MODIS deep blue aerosol products, Rev. Geophys., 50, RG3005, doi:10.1029/2012RG000388, 2012.

Mahowald, N., Albani, S., Engelstaedter, S., Winckler, G., and Goman, M.: Model insight into glacial-interglacial paleodust records, Quat. Sci. Rev., 30, 832–854, 2011.

---

## Referee Comment (RC2) · Anonymous Referee #2 · 2 Jul 2019

This paper presents a very interesting dataset from a deep ice core extracted from El-brus in 2009. Several variables are measured and the record shows interesting trends in dust deposition in the Caucasus. The paper is generally well written and fits the aim and scope of ACP. I suggest it can be published after some minor revisions mainly related to the structure of the paper. In fact, often results and methods are mixed together. More details on the statistical data analysis should be added in the Methodology section. Herafter, my specific comments on the paper.

pg1-ln23 Please define the acronyms (PDO, SOI)

pg1-ln25 Further information regarding anthropogenic activities will be helpful here

pg2-ln4 Please add some references and examples regarding the usage of satellite

data for dust monitoring

pg2-ln15 please correct: "proxy data are fundamental"

pg2-ln26 please correct: "the caucasus is a natural trap"

pg2-ln28-29 not clear, please rephrase

pg3-ln18-21 This is more an "abstract-like" sentence. I don't think is necessary to introduce the results in this Section

pg4-ln1 here is not clear how you separate winters and summers, please add details

pg4-ln10 "Dating the ice core"

pg4-ln15 Remove "Susanne" from the reference

Section 3.3 I think that this section should be reshaped. Important methodological descriptions are introduced here, but they are mixed up with the results. This does not help the reader. I suggest to move all the methodological information to Section 3.1 and to create a new paragraph in Section 4 where the authors introduce the structure of their results.

p5-ln31 here other impacts of dust in the cryosphere should be mentioned. For example the impact on snow and ice melting throung snow-albedo feedback (see. Gabbi et al. 2015 Cryosph.; Di Mauro et al. 2019 Cryosph.)

pg6-ln13 are there other studies focusing on these peaks? If yes, please add some references.

pg6-17 the trend analysis should be described in the Methods. Further details on the statistical analysis should be addedd. How the authors checked the significance of trends? They used a parametric or non-parametric methods?

pg6-ln28 here the authors may reference to recent large dust transports happened in spring 2018.

pg7-ln28 here the authors should briefly describe the possible impact of this increase of dust concentration. For example, earlier snow melt at lower altitude, (more) negative mass balance of glaciers, higher frequency of avalanches etc.

pg7-ln28 is this increase reflected in ERA-interim data? Please add details on it. ERA-interim data are referenced only in the caption of Figure 9. Please add details on how and why you used this dataset.

pg8 ln2: add a comma after Sahara.

pg8-ln12 add this in the methods. How it was calculated, from which data? SPEI 3 was the most correlated with Ca2+, you tried other indices? why the average of three months results in a higher correlation?

pg8-ln33 "number of days", of what?

pg9-ln12 not clear. Please rephrase.

pg9-ln22-23 these indices were never introduced. Acronyms are not defined. Add this information in the methods section. Describe why you selected those indices.

pg10-ln26 Add brackets to "Dai (2011)"

pg10-ln26 Define SST

References: De Angelis and Gaudichet 1991, De Angelis et al. 1997 and De Chatel 2014 are not listed in alphabetical order, please correct.

Figure 1: This map needs a legend. Colors are derived from hysplit backward trajectory but they are not clear to the reader, who cannot interpret them without a legend. I suggest to create a new figure with a clear evidence of possible dust sources. Furthermore, a map depicting the location of the drilling site on Elbrus could be useful.

Figure 2: describe in the caption what's in the different panels

Figure 4: delete "RAW DATA" from the legend. Please mark the zoom the in upper

panel.

Figure 7: please add the labels to both plots

Figure 10: is this a piece-wise regression? How the authors identified the break? This is important since the detection of the break strongly influences the correlation they show with SPEI. Here the authors show the Pearson's coefficient r, whereas in Figure 7 they show $R^2$. I suggest to use $R^2$ across the whole manuscript.

---

## Author Comment (AC1) · 9 Sep 2019

**Reply to reviewers' comments on "History of desert dust deposition recorded in the Elbrus ice core" We would like to thank both reviewers for their comments that help us to improve and clarify the manuscript.**

Please note that this is a companion paper of another manuscript submitted to ACP [https://www.atmos-chem-phys-discuss.net/acp-2019-402/](https://www.atmos-chem-phys-discuss.net/acp-2019-402/) Preunkert et al., "The Elbrus (Caucasus, Russia) ice core glaciochemistry to reconstruct anthropogenic emissions in central Europe: The case of sulfate."

Some additional text was added to this manuscript as suggested by the reviewer of the Preunkert et al. paper (see section 4.3). Figure 5 was changed and an additional Figure 7 was added following the recommendation of one of the reviewers as well.

**Reply to**

**Anonymous Referee #1**

The manuscript by Kutuzov et al. presents a calcium-dust proxy record from an ice core drilled on Mt. Elbrus, Caucasus, spanning the time period 1774-2013 CE. The discussion on the dust proxy include separating the background signal and the main dust events, evaluating the frequency and amplitude of dust events, establishing a relation with the potential dust sources by means of analysis of atmospheric circulation patterns and climate indices. I found the dataset very interesting per se, which warrants publication in ACP. The discussion and interpretation of the record is quite detailed and includes very interesting findings, following an approach established by the same group of authors. However, I also found that one relevant issue, related to the almost 10-fold increase in snow accumulation with potential implications on the interpretation of the proxy record, is not discussed. Therefore I recommend a major review of the manuscript.

General comments

After reading the manuscript, I am not sure whether the trends mostly reflect changes in deposition/accumulation processes rather than changes in dust emissions. This aspect is not discussed in the manuscript, while I believe it is central for the interpretation of the record. I elaborate on this consideration in the lines below.

In Figure 1a one can be appreciate how the summer half-year thickness, expressed in meters of water equivalent, increases significantly since the beginning of the record. This implies increased snow accumulation, in addition to the expected ice compaction with depth. In fact the "companion" discussion paper by Preunkert et al., reports a "decrease of the net annual snow accumulation from 1.5 mwe (0.8 mwe in summer and 0.7 mwe in winter) near the surface to 0.18 mwe (0.15 mwe in summer and 0.03 mwe in winter) at 157 m depth". Therefore we see an almost 10-fold increase in snow accumulation rates along the core. The authors discussed the related potential issues in determining the the ability to detect the frequency of dust events; in order to overcome this issue,

they adopted a strategy with finer sampling in the bottom sections of the core. While this precaution is an effective measure to that aim, it does not respond to the issue of whether the increased accumulation rates reflect increased precipitation and wet scavenging, in other words a larger or more frequent sampling of the atmospheric dust loading during precipitation events. As a result, it cannot be safely concluded which effect primarily (or maybe both) determines the observed trends in the dust proxy. This kind of reasoning is partly grounded in the long-standing debate on whether for instance dust concentrations or deposition fluxes are a better proxy for atmospheric dust / dust variations (e.g. Fischer et al., 2007; Mahowald et al., 2011).

I recommend that these issues are throughly discussed in the manuscript, and the interpretations and conclusions weighted accordingly.

*Taken into account. We thank reviewer for these general comments. One important issue was raised by the reviewer. The significant change in the accumulation rate indeed may influence the results of ice cores interpretation. However, in this manuscript we do not show or discuss any accumulation changes. The presented figures and data show only the thickness of annual layers. In order obtain an accumulation rate the layer thickness must be corrected for the compression which occurred since it was deposited (e.g. Paterson and Waddington, 1984). This effect at Elbrus can be clearly seen at Figure 9 in (Mikhalenko et al., 2015) which shows the annual layer thickness and the Nye model fit. The monotonic decrease of annual layer thickness is an effect of layer thinning. A separate paper dedicated to the accumulation rate change in Elbrus should be submitted soon. The accumulation is calculated using the dating and available reference horizons together with depth age modelling. When accounted for the ice layer thinning the accumulation variations are within 20-30% and there is no linear trend in accumulation change over the whole period. Therefore we do not expect any significant influence of the accumulation change on Ca2+ concentrations giving the reasonable sampling resolution. The observed trends in Ca2+ concentration cannot be explained by the changes in accumulation.*

*We can see that this misunderstanding was due to inconsistent wording in figure captions and in the text of two companion manuscripts. We added paragraph to explain this issue.*

*"It show be noted that Fig. 3 show the thickness of layers and does not represent the linear change in accumulation rate. In order obtain an accumulation rate the layer thickness must be corrected for the compression which occurred since it was deposited (e.g. Paterson and Waddington, 1984) which is out of the scope of this paper. "*

Specific comments

p. 2 / lines 3-4: Please explain how

*Taken into account. Text revised. "The discrepancies between models are partly explained by very limited observations of dust variability over the past and therefore limited possibilities to evaluate the model's reproducibility of the dust cycle."*

2/4-5: please provide some references

*Done. References added (e.g. Gautam et al., 2009; Chudnovsky et al., 2017; Li and Sokolik, 2018).*

2/8: it would seem more precise to say that many of the archives reported in the cited manuscript show a doubling of the respective dust signals

*Taken into account. Text revised.*

2/9-13: Given the level of detail reported here with reference to the cited paper, it may be worth reporting other studies as well (e.g. Ginoux et al., 2012; older papers assessing the issue at the global level)

*Taken into account. Text revised, reference added.*

2/25: Two references are listed as Kutuzov et al. 2015 a,b. Please delete the nonrelevant one. In addition, remove from the reference list the discussion paper (Kutuzov et al., 2013).

*Done*

3/9: Make sure that the special character is properly displayed. In addition, in the legends of Figures 1 and 2, a resolution of 0.5 x 0.5 degree is reported. Which one is correct?

*Both the resolutions are correct. The first one is related to NCEP/NCAR Reanalysis, the second one simply show a resolution with which the Figures 1 and 2 were produced.*

3/12: Rather than aerosols, the HYSPLIT analysis reported here shows that "Elbrus glaciers receive AIR MASSES from sources . . ."

*Text revised*

3/14-21: Please clarify whether the density plots in Figure 2 are based solely on the the back-trajectories passing close to the ground (and what about Figure 1?). In addition, please explain how did you define the well-mixed boundary layer.

*Text revised. "Density plots were calculated only for 10 day backward trajectories which descended below mixed layer depth. The depth is calculated by HYSPLIT_4 (using NCEP/NCAR Reanalysis data) for each point of backward trajectory as the height of the first exceeding of potential air temperature over surface air temperature by 2 K in the point (Draxler & Hess, 1998)."*

3/25: Please report the geographical coordinates

*Done*

3/26: ranged from . . . AT 10 m depth . . .

*Done*

3/30: Could you report in a few words the main aspects of that methodology?

*Done. "Cores were subsampled and decontaminated at -15°C using the pre-cleaned electric plane tool methodology described in (Preunkert & Legrand, 2013). In brief, in a first step, ice samples were cut with a band saw. After that, all surfaces of the cut samples were cleaned under a clean air bench*

*by using a pre-cleaned electric plane tool over which the ice was slid. To control the decontamination efficiency process blank ice samples, consisting of ultrapure frozen MilliQ water were preceded regularly."*

3/30-31: Please specify whether the sampling is continuous along the core

*Done. Text revised.*

4/8-9: Define what is meant by decontamination blank

*Done. Text revised. "To control the decontamination efficiency process blank ice samples, consisting of ultrapure frozen MilliQ water were preceded regularly."*

4/18-19: Please rephrase

*Done. Text slightly revised*

5/7-8: Please report briefly the methodology of the cited paper, i.e. how one can identify dust events on the basis of Ca2+ and acidity records.

*The section with this part of the text was removed. This criteria is explained further in the manuscript in section 4.1.*

5/20-22: It would be interesting to estimate/report the uncertainty arising from this assumption, and propagate it to the dust proxy.

*Rejected. The sea salt Ca+2 fraction of the summer Ca2+ concentration is 1±0.7%. We show that we can neglect its influence.*

5/22-25: Can you provide an estimate of how much this uncertainty could amount to?

*Rejected. We really cannot say more than what is stated. "That percentage is clearly an upper limit since, in precipitation deposited at continental free tropospheric sites (e.g. Legrand, 2002), Na+ is not only related to sea-salt due to the presence of leachable sodium in alumino-silicate particles but also Na+ from halide evaporates present in the deserts".*

*This upper limit (1 and 1.5%) is low enough to completely neglect the sea-salt contribution to the calcium level.*

5/31 - 6/3: It is not clear what you mean by "disturb" in both sentences. Please rephrase.

*Taken into account. Text revised.*

6/8: What do yo mean by "warm periods"? Warm years/decades? Warm seasons?

*Warm seasons. Text revised.*

References

Fischer, H., Siggaard-Andersen, M. L., Ruth, U., Rothlisberger, R., and Wolff, E. W.: Glacial/Interglacial changes in mineral dust and sea-salt records in polar ice cores: sources, transport, and deposition, Rev. Geophys., 45, RG1002, doi:10.1029/2005RG000192, 2007.

Ginoux, P., Prospero, J., Gill, T., Hsu, N., and Zhao, M.: Global-scale attribution of anthropogenic and natural dust sources and their emission rates based on MODIS deep blue aerosol products, Rev. Geophys., 50, RG3005, doi:10.1029/2012RG000388, 2012.

Mahowald, N., Albani, S., Engelstaedter, S., Winckler, G., and Goman, M.: Model insight into glacial-interglacial paleodust records, Quat. Sci. Rev., 30, 832–854, 2011

*References*

*Paterson, W. S. B. and Waddington, E. D.: Past precipitation rates derived from ice core measurements: methods and data analysis, Rev. Geophys., 22, 123–130, 1984*

*Mikhalenko, V., Sokratov, S., Kutuzov, S., Ginot, P., Legrand, M., Preunkert, S., Lavrentiev, I., Kozachek, A., Ekaykin, A., Faïn, X., Lim, S., Schotterer, U., Lipenkov, V., and Toropov, P.: Investigation of a deep ice core from the Elbrus western plateau, the Caucasus, Russia, The Cryosphere, 9, 2253-2270, https://doi.org/10.5194/tc-9-2253-2015, 2015.*

---

## Author Comment (AC2) · 9 Sep 2019

**Reply to reviewers' comments on "History of desert dust deposition recorded in the Elbrus ice core" We would like to thank both reviewers for their comments that help us to improve and clarify the manuscript.**

Please note that this is a companion paper of another manuscript submitted to ACP [https://www.atmos-chem-phys-discuss.net/acp-2019-402/](https://www.atmos-chem-phys-discuss.net/acp-2019-402/) Preunkert et al., "The Elbrus (Caucasus, Russia) ice core glaciochemistry to reconstruct anthropogenic emissions in central Europe: The case of sulfate."

Some additional text was added to this manuscript as suggested by the reviewer of the Preunkert et al. paper (see section 4.3). Figure 5 was changed following the recommendation of one of the reviewers as well.

**Reply to**

**Anonymous Referee #2 (RC2)

This paper presents a very interesting dataset from a deep ice core extracted from Elbrus in 2009. Several variables are measured and the record shows interesting trends in dust deposition in the Caucasus. The paper is generally well written and fits the aim and scope of ACP. I suggest it can be published after some minor revisions mainly related to the structure of the paper. In fact, often results and methods are mixed together. More details on the statistical data analysis should be added in the Methodology section. Hereafter, my specific comments on the paper.

pg1-ln23 Please define the acronyms (PDO, SOI)

*Done*

pg1-ln25 Further information regarding anthropogenic activities will be helpful here

*If we understand correctly the reviewer asked to provide more details on whether the anthropogenic activity contributed to the increasing trend of Ca2+. Text was revised. Sentence added "It was shown that the increase of $Ca^{2+}$ concentration in the ice core cannot be attributed to human activities, such as coal combustion and cement production."*

pg2-ln4 Please add some references and examples regarding the usage of satellite data for dust monitoring

*references added (e.g. Chudnovsky et al., 2017; Gautam et al., 2009; Li and Sokolik, 2018)*

pg2-ln15 please correct: "proxy data are fundamental"

*done*

pg2-ln26 please correct: "the caucasus is a natural trap"

*done*

pg2-ln28-29 not clear, please rephrase

*Changed to «The absence of melt water infiltration near the summit of Elbrus ensures the preservation of a climatic record in an ice core.»*

pg3-ln18-21 This is more an "abstract-like" sentence. I don't think is necessary to introduce the results in this Section

*Accepted. Moved to section 4.1*

pg4-ln1 here is not clear how you separate winters and summers, please add details

*The details of ice core dating are provided in the next section. (3.2.). We moved the paragraph about the sampling resolution to this section.*

pg4-ln10 "Dating the ice core"

*Done*

pg4-ln15 Remove "Susanne" from the reference

*Done*

Section 3.3 I think that this section should be reshaped. Important methodological descriptions are introduced here, but they are mixed up with the results. This does not help the reader. I suggest to move all the methodological information to Section 3.1 and to create a new paragraph in Section 4 where the authors introduce the structure of their results.

*Taken into account. We removed this section and moved part of the text to Section 4. The second paragraph was deleted.*

p5-ln31 here other impacts of dust in the cryosphere should be mentioned. For example the impact on snow and ice melting throung snow-albedo feedback (see. Gabbi et al. 2015 Cryosph.; Di Mauro et al. 2019 Cryosph.)

*Taken into account.*

*Following sentence was added: Deposition of light-absorbing impurities (in particular, black carbon and dust) plays an important role in changes of the snow and glaciers and may enhance the response of the mountain cryosphere to climate changes via snow-albedo feedbacks (Gabbi et al., 2015; Ginot et al., 2014; Mauro et al., 2017; Skiles et al., 2018)*

pg6-ln13 are there other studies focusing on these peaks? If yes, please add some references.

*We're not aware of other studies of these dust peaks in Caucasus. Comparison to other records are presented in section 4.5*

pg6-17 the trend analysis should be described in the Methods. Further details on the statistical analysis should be addedd. How the authors checked the significance of trends? They used a parametric or non-parametric methods?

*Taken into account. For correlation analysis we used Pearson's correlation and correlation significance was checked using a t-test. The t-test has been also used to quantify the significance of linear trends. A new section (3.3) on climate data analysis was added.*

pg6-ln28 here the authors may reference to recent large dust transports happened in spring 2018.

*Accepted. the sentence was added «Recent major long-range dust outbreak event over North Africa, eastern Mediterranean, and Caucasus occurred on 22 and 23 March 2018 (Solomos et al., 2018)».*

pg7-ln28 here the authors should briefly describe the possible impact of this increase of dust concentration. For example, earlier snow melt at lower altitude, (more) negative mass balance of glaciers, higher frequency of avalanches etc.

*Declined. This topic is out of the scope of this paper. We do not calculate and radiative forcing neither discussing potential impacts of dust on snow and glaciers. This needs a separate detailed study. Answering to the previous comment a sentence was added to the section 4.1. "Deposition of light-absorbing impurities (in particular, black carbon and dust) plays an important role in changes of the snow and glaciers and enhance the response of the mountain cryosphere to climate changes via snow-albedo feedbacks (Ginot et al., 2014; Gabbi et al., 2015; Di Mauro et al., 2017; Skiles et al., 2018)."*

pg7-ln28 is this increase reflected in ERA-interim data? Please add details on it. ERAinterim data are referenced only in the caption of Figure 9. Please add details on how and why you used this dataset.

*Done. Text revised. A new section (3.3) on climate data analysis was added.*

pg8 ln2: add a comma after Sahara.

*Done*

pg8-ln12 add this in the methods. How it was calculated, from which data? SPEI 3 was the most correlated with Ca2+, you tried other indices? why the average of three months results in a higher correlation?

*Text was revised. A paragraph was added to the new methodology section 3.3.*

*The Standardised Precipitation-Evapotranspiration Index (SPEI) was used as a drought proxy (Vicente-Serrano et al., 2010). It was showed that this index is preferable among the other drought indices to investigate the relation of dust emission to droughts (Achakulwisut et al., 2018). We considered SPEI calculated over 1, 2, 3, 6 and 12 month. The time series of SPEI were obtained by averaging over the regions of interest.*

*We only report the highest correlation which was found for the SPEI3. We can only speculate that the persistent drought for three months may result in a higher dust emission. But this is rather bold statement. All the drought indices show an increase in dryness of the regions. Which is our main message.*

pg8-ln33 "number of days", of what?

*Noted. The number of dust days*

pg9-ln12 not clear. Please rephrase.

*Done*

pg9-ln22-23 these indices were never introduced. Acronyms are not defined. Add this information in the methods section. Describe why you selected those indices.

*Section 3.3. was added to the methods. Description of the data used added.*

pg10-ln26 Add brackets to "Dai (2011)"

*Done*

pg10-ln26 Define SST

*Done*

References: De Angelis and Gaudichet 1991, De Angelis et al. 1997 and De Chatel 2014 are not listed in alphabetical order, please correct.

*Done*

Figure 1: This map needs a legend. Colors are derived from hysplit backward trajectory but they are not clear to the reader, who cannot interpret them without a legend. I suggest to create a new figure with a clear evidence of possible dust sources. Furthermore, a map depicting the location of the drilling site on Elbrus could be useful.

*Taken into account. Figure 1 was revised. We don't see the necessity in adding the addintional figure since the location of Elbrus is shown on Fig. 1.*

Figure 2: describe in the caption what's in the different panels

*Done*

Figure 4: delete "RAW DATA" from the legend. Please mark the zoom the in upper

*Done*

Figure 7: please add the labels to both plots

*Done*

Figure 10: is this a piece-wise regression? How the authors identified the break? This is important since the detection of the break strongly influences the correlation they show with SPEI. Here the authors show the Pearson's coefficient r, whereas in Figure 7 they show $R^2$. I suggest to use $R^2$ across the whole manuscript.
*Taken into account. SPEI data are available from 1901. The linear trend for the Ca2+ data was calculated for the same period.*

*We tend to decline the reviewers suggestion to use $R^2$ across the manuscript. Since we present the strength and the direction of the linear relationship between two time series the r is a common choice. r is a common measure of similarity between time series. On contrary $R^2$ is normally used to*

*show how well the model follows the data. On figure 7 we report the scatterplot and linear regression of two chemical species and so $R^2$ is commonly used for such plots. We do not see inconsistency here.*

References:

*Chudnovsky, A. A., Koutrakis, P., Kostinski, A., Proctor, S. P. and Garshick, E.: Spatial and temporal variability in desert dust and anthropogenic pollution in Iraq, 1997–2010, J. Air Waste Manag. Assoc., 67(1), 17–26, doi:10.1080/10962247.2016.1153528, 2017.*

*Gabbi, J., Huss, M., Bauder, A., Cao, F. and Schwikowski, M.: The impact of Saharan dust and black carbon on albedo and long-term mass balance of an Alpine glacier, Cryosphere, 9(4), 1385–1400, doi:10.5194/tc-9-1385-2015, 2015.*

*Gautam, R., Liu, Z., Singh, R. P. and Hsu, N. C.: Two contrasting dust-dominant periods over India observed from MODIS and CALIPSO data, Geophys. Res. Lett., 36(6), L06813, doi:10.1029/2008GL036967, 2009.*

*Ginot, P., Dumont, M., Lim, S., Patris, N., Taupin, J. D., Wagnon, P., Gilbert, A., Arnaud, Y., Marinoni, A., Bonasoni, P. and Laj, P.: A 10 year record of black carbon and dust from a Mera Peak ice core (Nepal): Variability and potential impact on melting of Himalayan glaciers, Cryosphere, 8(4), 1479–1496, doi:10.5194/tc-8-1479-2014, 2014.*

*Li, L. and Sokolik, I. N.: Analysis of dust aerosol retrievals using satellite data in Central Asia, Atmosphere (Basel)., 9(8), doi:10.3390/atmos9080288, 2018.*

*Mauro, B. Di, Baccolo, G., Garzonio, R., Giardino, C. and Massabò, D.: Impact of impurities and cryoconite on the optical properties of the Morteratsch Glacier ( Swiss Alps ), , 2393–2409, 2017.*

*Skiles, S. M., Flanner, M., Cook, J. M., Dumont, M. and Painter, T. H.: Radiative forcing by light-absorbing particles in snow, Nat. Clim. Chang., 8(11), 964–971, doi:10.1038/s41558-018-0296-5, 2018.*

*Solomos, S., Kalivitis, N., Mihalopoulos, N., Amiridis, V., Kouvarakis, G., Gkikas, A., Binietoglou, I., Tsekeri, A., Kazadzis, S., Kottas, M., Pradhan, Y., Proestakis, E., Nastos, P. and Marenco, F.: From Tropospheric Folding to Khamsin and Foehn Winds: How Atmospheric Dynamics Advanced a Record-Breaking Dust Episode in Crete, Atmosphere (Basel)., 9(7), 240, doi:10.3390/atmos9070240, 2018.*

---

## Author Response (AR2)

Dear Yves,

Please find below a revised version of the manuscript.

English language was checked and improved by a native speaker.

Kind regards,

Stanislav Kutuzov

[revised manuscript text omitted]